# The Thiol Group Reactivity and the Antioxidant Property of Human Serum Albumin Are Controlled by the Joint Action of Fatty Acids and Glucose Binding

**DOI:** 10.3390/ijms25042335

**Published:** 2024-02-16

**Authors:** Tamara Uzelac, Katarina Smiljanić, Marija Takić, Ivana Šarac, Gordana Oggiano, Milan Nikolić, Vesna Jovanović

**Affiliations:** 1Department of Biochemistry and Centre of Excellence for Molecular and Food Sciences, University of Belgrade—Faculty of Chemistry (UBFC), Studentski trg 12-16, 11158 Belgrade, Serbia; tamarauzelac31@gmail.com (T.U.); katarinas@chem.bg.ac.rs (K.S.); mnikolic@chem.bg.ac.rs (M.N.); 2Centre of Research Excellence in Nutrition and Metabolism, Group for Nutrition and Metabolism, National Institute of Republic of Serbia, Institute for Medical Research, University of Belgrade, Tadeuša Košćuškog 1, 11000 Belgrade, Serbia; marijapo2001@gmail.com (M.T.); ivanasarac@yahoo.com (I.Š.); g5petrovic@gmail.com (G.O.)

**Keywords:** antioxidant role, fatty acids, glucose, glycation, human serum albumin, thiol group content and reactivity

## Abstract

The binding of ubiquitous serum ligands (free fatty acids) to human serum albumin (HSA) or its glycation can affect thiol group reactivity, thus influencing its antioxidant activity. The effects of stearic acid (SA) and glucose binding on HSA structural changes and thiol group content and reactivity were monitored by fluoroscopy and the Ellman method during a 14-day incubation in molar ratios to HSA that mimic pathophysiological conditions. Upon incubation with 5 mM glucose, HSA glycation was the same as HSA without it, in three different HSA:SA molar ratios (HSA:SA-1:1-2-4). The protective effect of SA on the antioxidant property of HSA under different glucose regimes (5-10-20 mM) was significantly affected by molar ratios of HSA:SA. Thiol reactivity was fully restored with 5–20 mM glucose at a 1:1 HSA:SA ratio, while the highest thiol content recovery was in pathological glucose regimes at a 1:1 HSA:SA ratio. The SA affinity for HSA increased significantly (1.5- and 1.3-fold, *p* < 0.01) with 5 and 10 mM glucose compared to the control. These results deepen the knowledge about the possible regulation of the antioxidant role of HSA in diabetes and other pathophysiological conditions and enable the design of future HSA-drug studies which, in turn, is important for clinicians when designing information-based treatments.

## 1. Introduction

In blood plasma, human serum albumin (HSA) is the most abundant protein (35–55 g/L or 0.52–0.82 mM). It has an approximate half-life of 19 days and is degraded more effectively if denatured or structurally altered [1]. Moreover, in the intravascular space (~118 g), two-thirds (~242 g) of the total amount of HSA is present in the interstitial space of organs such as skin, gut, bone marrow, liver muscle, or in the body’s fluids and secretions such as cerebrospinal, pleural, sweat, tears and milk, while intracellularly it is present in lower concentrations [1,2]. Hepatocytes are the primary site of HSA synthesis, and approximately 13.9 g of HSA is secreted from the liver into the bloodstream each day. After HSA enters the bloodstream, it can translocate through the pores of sinusoidal or fenestrated capillaries in certain organs such as the liver, pancreas, small intestine, and bone marrow into the interstitial fluid or be actively transported through the continuous capillaries by seven membrane-associated albumin-binding receptors [1,3]. The lymphatic system transports HSA from the interstitial space back into the bloodstream, and in the course of its life, HSA makes around 28 “trips” in and out of the lymphatic system [1,2].

HSA is a versatile molecule and has a number of important biological functions. It is responsible for controlling fluid distribution between body compartments by maintaining oncotic pressure and blood pH [1]. HSA can bind and transport various endogenous molecules such as long-chain fatty acids (FAs), bilirubin, heme, thyroxine, bile acids, L-tryptophan (Trp), steroids, and vitamins that have a low solubility in water [1,2,4]. HSA also binds and transports different ions, including copper, zinc, calcium, cobalt, and iron [1,2,5]. In addition, it can bind numerous exogenous compounds, such as drugs, polyphenols, pesticides, and toxins, with an affinity that can influence their activity and half-life [1,2,6,7]. Interestingly, HSA is the most important extracellular antioxidant [1,2,8]. Its antioxidant activity can be manifested through several mechanisms, such as the elimination of free radical species by the free thiol group [9] and the binding of prooxidants (bilirubin, free copper, or iron) or oxidation-sensitive molecules (unsaturated FAs) [1,2,8]. Finally, HSA shows different (pseudo)-enzymatic properties such as esterase, enolase, glucuronidase, and peroxidase activity [10].

HSA is a single-chain globular protein composed of 585 amino acid residues arranged in three domains labeled I, II, and III, encompassing amino acids 1–195, 196–383, and 384–585, respectively [10]. Each domain is further divided into subdomains A and B. The HSA secondary structure consists of α-helix (67%) and random coils, without β-sheet. Each domain has 10 helices, six in subdomain A and four in subdomain B, connected by a long loop [2,11]. The specific spatial arrangement of the domains and subdomains in HSA results in the heart-shaped three-dimensional structure of the molecule, in which the gap separates the subdomains IA-IB-IIA on one side from the subdomains IIB-IIIA-IIIB on the other side of the molecule [1,2].

This unique shape of the HSA molecule creates multiple binding sites for different endogenous and exogenous molecules. There are nine FA binding sites named FA1 through FA9, of which seven FA binding sites (FA1–FA7) are found in all subdomains of HSA. The FA8 and FA9 sites are considered supplementary binding sites because they are positioned in the gap of the HSA molecule and are only occupied in the presence of short-chain FAs or increased FA levels in the blood [12,13]. Under normal physiological conditions, the molar ratio of HSA:FA is between 1:0.2 and 1:2. In contrast, the ratio can increase up to 1:6 in some pathological conditions such as diabetes, non-alcoholic fatty liver, and cardiovascular diseases, as well as during starvation or physical activity [1,2,14].

In addition to seven common FA sites, there are two main versatile ligand-binding sites: Sudlow sites I and II, which have a high affinity for different molecules. Sudlow site I (warfarin-azapropazone binding site) is the major drugs binding pocket located in subdomain IIA, which can be further divided into three non-overlapping subsites for salicylic acid at the bottom of Sudlow site I, indomethacin in the middle of Sudlow site I, and 3-azido-3-deoxythymidine near the entrance of Sudlow site I [15]. Sudlow II (indole-benzodiazepine binding site) is located in subdomain IIIA. Binding site FA7 partially overlaps with the Sudlow I site, while binding sites FA3 and FA4 overlap with the Sudlow II site [1,14]. Due to the flexible structure of HSA, there are competitive and/or cooperative interactions during the simultaneous binding of different ligands to HSA [6,7,15].

Interestingly, Wang et al. [15] found two different forms of glucose molecules: one in pyranose form and another in the open linear form located in Sudlow site I. The cyclic form of glucose was found near the bottom of the pocket, while the open form was located at the entrance of the pocket. Additionally, Wang et al. [15] found two molecules of fructose located in the same pocket of HSA crystals, while a molecular dynamics simulations study revealed the possible presence of galactose molecules in the same pocket, as well as that Sudlow I site prefers glucose over galactose [16]. So far, the physiological significance of binding highly water-soluble molecules such as glucose to the Sudlow I site has only been discussed from the aspect of glycation of lysine (Lys)195 of HSA [15,16,17].

As a major extracellular antioxidant, HSA has one free thiol group, cysteine (Cys)34, (HSA-SH) that is 70% in the reduced state and contributes up to 370–474 µM of the total plasma thiols (400–600 µM) under normal physiological conditions [1,18]. The reversible oxidation of the HSA-SH (25%) to disulfide with low molecular weight thiols, mainly Cys, is the main mechanism by which this group protects the body from reactive oxygen species (ROS). In addition, the HSA-SH (1–2%) can be irreversibly oxidized to sulfinic and sulfonic acids [1,19]. The efficacy of the HSA-SH in protecting the body against ROS depends on its availability and reactivity [5,19].

It was shown that the binding of various FAs and ligands to HSA, such as polyphenols, drugs, and metals, alters the availability and reactivity of the HSA-SH [4,5,6,7,20]. ROS, glucose, and other carbonyl compounds can react directly with the HSA-SH or indirectly with other groups in side chains of amino acids in HSA, changing the thiol group content or reactivity [4,21], and/or the affinity of HSA for different ligands [22,23,24]. Due to these mutual effects, HSA decreases its affinity for FAs, drugs, and metal ions or alters its redox state in conditions such as aging, diabetes mellitus, renal, cardiac, and hepatic dysfunction, or physical activity [5,21,22,24,25,26,27]. Accordingly, the HSA molecule is susceptible to environmental changes and is very adaptable to its physiological functions such as transport and antioxidant action in pathological conditions.

So far, the significance of interplay between the level of HSA glycation, the transport of FAs by HSA, and vascular disorders has not been clarifiedin conditions such as hyperglycemia and hyperlipidemia. Lautenslager et al. [23] found that fatty acids can decrease the ability of HSA to undergo glycation by glucose. Also, it is known that glycation of HSA can induce conformational and structural changes of HSA that affect exogenous ligand binding [22,24] as well as functional properties of HSA, which may have important repercussions in lipid transport and atherogenesis [23]. Szkudlarek et al. (2023) [28] confirmed that the glycation of HSA in the presence of glucose and palmitic acid (PA) causes changes in both HSA and its glycated tertiary structures, respectively. Moreover, PA, at ratios of 1.5:1 and 3:1 with glycated HSA, does not exhibit inhibition of advanced glycation end products (AGEs) formation. This study could point out that the structural changes of glycated HSA are important for pharmacological treatment planning because the type of interaction between the drugs and their primary transporter such as HSA may be altered as the disease progresses or in the elderly [28].

However, due to its importance in the metabolism and transport of FAs and the antioxidant protection of other molecules present in the extracellular space, it is more likely that HSA has some intrinsic mechanisms to maintain these functions. Considering that under normal physiological conditions, HSA binds up to two molecules of FAs [1,2], two molecules of glucose are located in Sudlow site I [15,16,17], and the glucose concentration is 3.9–7.8 mM [29], we hypothesized that both FAs and glucose may be involved in the maintenance of HSA functions under pathophysiological conditions.

To our knowledge, there are no data in the literature on the possible effects of the joint action of FAs and glucose in molar ratios to HSA, relevant to pathophysiological conditions on the HSA-SH content and reactivity, which are the main guardians of antioxidant homeostasis in intra/extravascular spaces. Therefore, the main aims of this study were: (1) to reveal how simultaneous presence of stearic acid (SA) and glucose in pathophysiological concentrations during in vitro glycation of HSA affects HSA-SH reactivity and content; (2) to find whether the joint action (synergy) of SA and glucose could affect the structural changes of HSA and formation of AGEs during glycation; (3) to assess if there is a tight connection between HSA functions and ubiquitous ligands (FAs and glucose) on their possible synergistic contribution in different vascular disorders (hyperglycemia, hyperlipidemia, etc.). The structural changes of HSA (molar ratios HSA:SA 1:0-1-2-4) caused by glycation (5-10-20 mM glucose) were monitored during a 14-day incubation at 37 °C by determining the intrinsic fluorescence of Trp214 and the formation of AGEs; HSA-SH content and reactivity by the Ellman method. In addition, the effects of different concentrations of glucose (5, 10, and 20 mM) on the binding of SA to HSA were investigated by determining the Stern–Volmer constants (Ksv).

## 2. Results

### 2.1. The Structural Changes of HSA during Glycation

The intrinsic fluorescence of HSA molecules and the fluorescent properties of AGEs were used to monitor the glycation-induced structural changes of HSA. Mixtures of HSA and SA were prepared at different HSA:SA molar ratios (1:0, 1:1, 1:2, and 1:4) and incubated in the absence (control) and presence of 5, 10, and 20 mM glucose for 14 days at 37 °C. The fluorescence spectra were recorded before (0 days) and after 4, 7, 10, and 14 days at 37 °C.

In all mixtures, the peak at 340 nm, which mainly represents Trp214 fluorescence, was about 5–14% lower when comparing the 14-day incubation with day 0 (Appendix A). A gradual decrease in peak intensity with the progression of incubation time was observed for the HSA:SA 1:0 and 1:2 molar ratios with 5 mM glucose and for HSA:SA 1:0, 1:2, and 1:4 mixtures with 10 mM glucose, while for other samples the trends of changes in peak intensity were irregular (Appendix A). These results indicate that during incubation the Trp214 environment was constantly changing and this was caused by glycation of HSA. The level of changes was affected by both the HSA:SA ratio and the concentration of glucose.

#### 2.1.1. The Effects of Glucose Concentrations on Trp214 Fluorescence

The increasing concentration of glucose led to a gradual decrease in the fluorescence intensity in the HSA:SA 1:0 and 1:2 samples after 14 days of incubation, and it was more manifested in sample 1:0 (Figure 1a,c). The trend of fluorescence intensity decrease was in line with previously published data [23,28], but it was not so profound. Interestingly, there was no fluorescence intensity change compared to the HSA control when the HSA:SA molar ratios were 1:1 and 1:4, and the glucose concentrations were 10 and 5 mM, respectively (Figure 1b,d).

#### 2.1.2. The Effects of the Molar Ratios of HSA:SA on Trp214 Fluorescence

All samples of HSA with bound SA have a higher fluorescence intensity than those without SA in the presence of all three glucose concentrations after 14 days of incubation (Figure 2). The increasing molar ratio of HSA:SA 1:0 to 1:4 led to a gradual increase in the fluorescence intensity only in the presence of 5 mM glucose (Figure 2a). When glucose concentrations were in the pathological range of 10 and 20 mM, the fluorescence intensities were the same for HSA:SA 1:2 and 1:4, and all mixtures with SA bound to HSA, respectively (Figure 2). These results are unique since there is no study that has followed mutual binding effects of different HSA:SA ratios in the presence of various glucose concentrations, set so as to reflect different molar ratios to HSA, covering at the same time healthy and disease state, as elaborated in this study.

#### 2.1.3. The Effects of Glucose Concentration on the Formation of AGEs

The fluorescence intensity at 430 nm, which denotes emission from AGEs, gradually increases with a progression of incubation time in all HSA incubation mixtures containing SA and in the presence of glucose (Appendix A).

After 14 days of incubation at 37 °C, the most profound effect of increasing glucose concentration on fluorescence intensity was observed at the HSA:SA molar ratio of 1:0 compared to the control (Figure 3a), and increasing the concentration from 10 to 20 mM had no effect on further AGEs formation. The fluorescence intensities for all three HSA:SA molar ratios in the presence of 5 mM glucose were at the same level as a corresponding control. Interestingly, the fluorescence intensities were slightly elevated in the presence of 10 and 20 mM glucose compared to the control for the HSA:SA 1:4 sample, mimicking the pathological conditions (Figure 3b,d). These results indicated that the formation of AGEs due to glycation by glucose in the normal physiological concentration (5 mM) was prevented when SA was bound to HSA. The protective effect of bound SA molecules to HSA (HSA:SA 1:4) was especially noticed when glucose was present in the pathological concentrations (10 and 20 mM). Similarly to previous results, these could not be placed into the literature context, since no similar experiments were published.

#### 2.1.4. The Effects of the Molar Ratios of HSA:SA on the Formation of AGEs

At all three glucose concentrations tested, the fluorescence intensities of AGEs were lower in HSA with bound SA than in HSA without it, except for the HSA:SA 1:1 sample in the presence of 20 mM glucose, where they were the same after 14 days of incubation (Figure 4). Gradually increasing the HSA:SA ratio from 1:1 to 1:4 resulted in a decrease in fluorescence intensity. At 5 mM glucose, the lowest intensity was achieved with two SA molecules bound to HSA and further increase in SA molar ratio to HSA had no effect (Figure 4a). In contrast to this, with the glucose concentration in the pathological range (20 mM), at least two molecules of SA were required for protection, and a further increase in the number of SA molecules enhanced this effect (Figure 4c). These results confirmed that the level of HSA glycation was determined by the joint action of FAs and glucose. According to this, the HSA molecule is capable of sensing the changes in FA and glucose concentrations in the intra/extravascular spaces. This sensitivity of HSA is important in conditions when the concentrations of FAs and glucose are rapidly changed such as before and after meals, during intensive physical activity, or diabetes.

### 2.2. Changes in HSA-SH Content during 14 Days of Incubation

The effect of structural changes of HSA caused by glycation on the HSA-SH group content was followed during 14 days of incubation in all prepared mixtures. Before (0 days) and 1–4, 6, 7, 10, and 14 days of incubation, HSA-SH content was determined by the Ellman method. The values obtained for the statistical comparisons between time points of all incubation mixtures using one-way ANOVA are given in Appendix A.

#### 2.2.1. The Effects of Glucose Concentration on HSA-SH Content

In all incubation mixtures with glucose, a decrease can be noticed in HSA-SH content during the first four days; after that, it is increased except for the molar ratio HSA:SA 1:4, where the decrease is continued up to 10 days of incubation (Figure 5). After 4 days of incubation, the smallest and highest decrease in HSA-SH content was obtained for the samples HSA:SA 1:0 with 10 mM glucose (6.1%) and HSA:SA 1:2 with 20 mM glucose (25.2%), respectively. After 14 days of incubation in the sample HSA:SA 1:0 with 20 mM glucose, the HSA-SH content was returned to the level before incubation, while in other mixtures with glucose, the values were 2.5–19.9% lower compared to 0 days. Interestingly, the mean values of the HSA-SH content obtained for the mixtures in the presence of glucose in selected time points were mainly higher than the corresponding control, and this was the most prominent in the sample HSA:SA 1:0 (Figure 5a). The decreasing trend of HSA-SH content during glycation was in line with previously published data [5,30]. These results point out that HSA-SH content is also influenced by the joint action of FAs and glucose and this was in accordance with the results of Szkudlarek et al. [28]. It is surprising that pathological glucose concentrations (10 and 20 mM) show a protective role in terms of antioxidant capacity, since the recovery of HSA-SH was evident, especially in HSA:SA 1:1 and 1:0 ratios.

#### 2.2.2. The Effects of the Molar Ratios of HSA:SA on HSA:SH Content

For all three glucose concentrations tested, a gradual decrease in HSA-SH content was observed after 14 days of incubation with an increase in the HSA:SA ratio from 1:1 to 1:4 (Figure 6). At most of the time points, the mean values of HSA-SH content obtained for HSA:SA 1:0 were higher than the mixtures in the presence of SA for each glucose concentration, which was most evident at 20 mM glucose (Figure 6c). At normal glucose concentration (5 mM), the recovery of HSA-SH content increased with a decrease in HSA:SA molar ratio from 6 to 14 days of incubation, and the same trend was noticed at 20 mM glucose from 7 to 14 days of incubation. According to these results, it can be concluded that increasing the HSA:SA ratio had a negative effect on the recovery of HSA-SH content. This could explain the reason for the observed lower HSA-SH content in the pathological conditions with elevated free FAs and glucose concentrations, such as type II diabetes or insulin resistance [21,27,31], in contrast to type I diabetes with reduced free FAs and the higher HSA-SH content [27,31].

### 2.3. Changes in HSA-SH Reactivity during Glycation

The influence of glycation on the changes in HSA-SH reactivity was determined before (0 days) and 1–4, 6, 7, 10, and 14 days of incubation in all incubation mixtures. Determination of the pseudo-first-order constant (k^−1^) for the reaction of HSA-SH with 5,5′-dithiobis-(2-nitrobenzoic acid) (DTNB) was used to estimate HSA-SH reactivity. The statistical comparisons between samples in different time points are given in Appendix A.

#### 2.3.1. The Effects of Glucose Concentration on the HSA-SH Reactivity

The shape of curves obtained for the HSA-SH reactivity in all mixtures in the presence of glucose was similar to those obtained for the HSA-SH content. Initially, a decrease in reactivity was followed by an increase (Figure 5 and Figure 7). The recovery of HSA-SH reactivity in the samples HSA:SA 1:1 with all glucose concentrations and HSA:SA 1:0 with 20 mM glucose was 100% (Figure 7a,b), while in the other mixtures, it was from 65 to 92%, upon 14-day incubation. Moreover, the HSA-SH reactivity was higher in the mixtures in the presence of glucose compared to the control, except for the mixtures prepared with HSA:SA 1:4. To our knowledge, this was the first study where the HSA-SH reactivity was followed during glycation. According to these results, it can be concluded that the HSA molecule has the ability to recover HSA-SH reactivity. The recovery level of HSA-SH reactivity was dependent on the HSA:SH molar ratio and glucose concentration. The presence of glucose in any concentrations showed a positive effect while increasing the HSA:SA ratio showed a negative effect. The capability of HSA to recover its HSA-SH reactivity has physiological importance because HSA as a major antioxidant has a role in protecting other proteins during oxidative stress.

#### 2.3.2. The Effects of the Molar Ratios of HSA:SA on the HSA-SH Reactivity

Gradual decrease in k^−1^ values with an increase in the HSA:SA ratio from 1:1 to 1:4 were obtained for 10 mM glucose (Figure 8b), while at 5 and 20 mM glucose, statistically significant higher values were shown in HSA:SA 1:1 ratio, upon 14-day incubation (Figure 8). At selected time points from 6 to 14 days of incubation in all three glucose concentrations, k^−1^ values for HSA:SA 1:0 and 1:1 were higher or equal compared to the values obtained for HSA:SA 1:2 and 1:4 (Figure 8). These results further confirmed that the increasing number of FAs bound to HSA prevents recovery of HSA-SH reactivity and its antioxidative role. This means that oxidative stress could be more pronounced in pathological conditions where the elevation of glucose and free FAs occurred, such as type II diabetes [21,27].

### 2.4. The Influence of Glucose on the Quenching Constant of HSA Trp214 with SA

As the mutual effects of FAs and glucose were demonstrated in all previous experiments, the influence of different glucose concentrations on the binding affinities of SA to HSA was investigated in the following experiment. Stern–Volmer constants (Ksv) of fluorescence quenching with SA in the absence (control) or the presence of 5, 10, and 20 mM glucose were determined at 37 °C (Table 1).

In the presence of 5 and 10 mM glucose, obtained values of Ksv were 1.5 and 1.3 times higher than the control (1.26 ± 0.03 × 10^4^), while for 20 mM glucose, the value was equal to the control. Hence, the glucose presence in the mixture with FA-free HSA (control) increased the Ksv constant in a decreasing concentration-dependent manner. In the preliminary experiment (data not shown) a gradual decrease in fluorescence intensity of FA-free HSA was obtained at HSA:SA ratios up to 1:4. Further increase in HSA:SA ratio led to transient increases of fluorescence intensity, while further addition of SA resulted in the fluorescence decrease. This fluctuation of fluorescence intensity was probably caused by the binding of SA molecules to the new FAs binding sites in the HSA molecule with a lower affinity to FAs. Because of this, the fluorescence intensities obtained for HSA:SA molar ratios from 1:0 to 1:4 were selected for the calculation of Ksv. The quenching fluorescence of excited fluorophores (Trp214 is in the Sudlow I site) of FA-free HSA with SA in the pathophysiological range of HSA:SA ratios could be caused by the joint action of the glucose molecules present in the Sudlow I site [15,16,17] and SA present in one of two FAs binding site in HSA with the highest affinity [2]. These high-affinity FA binding sites (FA5 and FA4) are located within domain III, of which FA4 is in the Sudlow II site.

## 3. Discussion

In normal physiological conditions, the HSA molecule bears 0.2 to 2 molecules of FAs [1,2] and is surrounded by the extracellular liquid in which the concentration of glucose is between 3.8 and 7.8 mM. We assumed that both ubiquitous plasma ligands are important in maintaining HSA structure and function in health, as well as during illness. To check this hypothesis, we followed the structural changes of HSA by the determination of intrinsic fluorescence of HSA molecules and AGEs, and HSA-SH content and reactivity during 14 days of glycation of HSA molecules under conditions that mimic pathophysiological concentration of glucose (5, 10, and 20 mM) and the molar ratios of HSA:SA 1:0, 1:1, 1:2, and 1:4.

Following the intrinsic fluorescence of Trp214 during 14 days of HSA incubation in the presence of SA and glucose (Appendix A) provided helpful information about how HSA structure was changing. The obtained decreases of Trp214 fluorescence intensity in all incubation mixtures after 14 days were moderate compared to the baseline fluorescence, and it was in line with the results of Szkudlarek et al. [24]. In both studies, glycation of HSA was followed in the presence of FAs. The other studies used a non-physiological molar ratio of HSA:glucose for glycation in the absence of FAs, and the decrease in Trp214 fluorescence was very profound [32]. HSA can be substantially modified by oxidative or carbonyl stress if the stressor concentrations are significantly higher than in the physiological conditions or in an incubation period longer than the half-life of HSA [21,22,32]. However, the mild modification of HSA structure was found in in vivo or in vitro studies when the concentration of stressors was in the pathophysiological range [24,27,33]. For example, after 21 days of incubation, with 0.63 mM HSA in the presence of 15 mM glucose at pH 7.4 and 37 °C, only one mol hexose/mol HSA was found [33], pointing to some intrinsic mechanisms by which HSA maintains its functions (antioxidative protection and substance carrier). These decreases in fluorescence intensities were not gradual in all samples during incubation time (Appendix A), and were influenced by both the HSA:SA molar ratios and the glucose concentrations. Because the small differences in the fluorescence intensities were obtained for different glucose concentrations (Figure 1) and the molar ratios of HSA:SA (Figure 2), the joint effect of glucose and FAs on structural changes of HSA during glycation was not so obvious.

In further analyses in which the effect of different glucose concentrations and HSA:SA molar ratios was investigated on the formation of AGEs and HSA-SH content and reactivity during 14 days of incubation, we obtained relevant proof of the joint action of FAs and glucose in maintaining the antioxidant role of HSA. We found that the formation of AGEs was at the same level as a corresponding control when HSA was incubated with SA (up to four molecules of SA per HSA) and 5 mM glucose after 14 days (Figure 3b,d), meaning that SA binding reduced AGEs formation, exerting a suppression of AGEs formation during glucose binding. This trend of suppression of AGE formation almost to the control level at the glucose pathological concentration range (10 and 20 mM), was achieved by increasing the number of bound SA to HSA (Figure 3 and Figure 4). For example, the peak intensities for HSA:SA 1:4 with 20 mM glucose and HSA:SA 1:1 with 5 mM were the same (Figure 4a,c). These results are in accordance with the study by Lautenslanger et al. [23], in which was proved that a physiological mixture or individual FAs in the molar ratio HSA:FAs 1:1 had a protective role in the formation of Amadori adducts during 7 days of incubation in the presence of 20 mM glucose. To our knowledge, our study was the first in which joint effects of glucose in different pathophysiological concentrations and HSA:FAs molar ratios were investigated on the HSA-SH content and reactivity during incubation, which corresponds to the actual half-life of HSA in circulation (12–21 days) [33].

So far, numerous in vivo studies have proved that the content of HSA-SH groups decreased in different pathological conditions compared to the control subjects or those who received some treatment [27,34,35,36,37]. Contrary to this, there are a few in vivo studies in which HSA-SH group content changes were followed in real time [26], while there are many in vitro studies [5,30].

Our in vitro study showed that the HSA-SH content could be recovered to some extent after initially decreasing during the first four days of incubation and the recovery was influenced by the joint action of FAs and glucose (Figure 5). These results are in accordance with the results of Szkudlarek [28], where the content of the HSA-SH group was higher in the presence of palmitic acid than in its absence after 21 days of HSA glycation with a glucose/fructose mixture. In our study, the highest recoveries were obtained for the mixtures HSA:SA 1:1 with 5 and 10 mM glucose and HSA:SA 1:0 with a 20 mM concentration of glucose (Figure 5). Increasing the number of FAs bound to HSA to two and four had a negative effect on the recovery of HSA-SH content (Figure 6). These results are physiologically relevant because they can explain the observed different HSA-SH content in patients with type 1 and 2 diabetes mellitus [27]. Individuals with type 1 diabetes mellitus with poor glycemic control have reduced free FA levels compared to healthy subjects [38]. This pathological condition was mimicked in our mixtures HSA:FAs 1:0 with 10 and 20 mM glucose (Figure 5a). The content of the HSA-SH group in these patients was the same as in the control group because recovery of HSA-SH is the highest. On the contrary, in patients with type 2 diabetes mellitus, the HSA-SH content was lower than in the control group [27]. Elevated free FAs present in the plasma of patients with type 2 diabetes or insulin resistance [31] and higher glucose concentrations could lead to less effective recovery of the HSA-SH content in these individuals during the half-life of HSA [21,27]. This pathological condition was mimicked in our mixtures HSA:FAs 1:2 and 1:4 with 10 and 20 mM glucose where the recovery of HSA-SH content was the lowest (Figure 6).

To our knowledge, the HSA-SH reactivity was analyzed only in several in vivo studies [7,39], and there are no published data in which the time course of HSA-SH reactivity was analyzed under physiological conditions. The limitation of this analysis is the prior isolation of HSA from serum. Affinity chromatography with Cibacron Blue Sephorose is a commonly used method for isolating HSA from serum or plasma [21]. Because it is time-consuming and selective to the HSA molecules that bear saturated FAs [39], we developed a rapid two-step ammonium-sulfate precipitation method for isolating HSA [40]. This method was used for the isolation of HSA from the sera of 10 runners before and after a half-marathon race. We obtained a transient increase in HSA-SH group reactivity and the molar ratio of HSA:FAs (1:2.34) and (1:1.27) 15 and 60 min after the race, respectively. After 24 h of the race, HSA-SH group reactivity and HSA/FAs ratio (1:0.84) were returned to the baseline values. According to the results of that in vivo study (manuscript in preparation) and the results obtained for this in vitro study, maintaining the reactivity of HSA-SH is significant. In this study, we obtained decreased reactivity of the HSA-SH group during the first three days of incubation and, after that, again increased in most of the tested mixtures (Figure 7). When the molar ratio HSA:SA was 1:1 for all tested glucose concentrations, after 14 days of incubation, the reactivity of HSA-SH returned to the baseline values (Figure 7b). Interestingly, decreased HSA-SH reactivity was more prominent in the mixture without glucose for the HSA:SA molar ratios of 1:0, 1:1, and 1:2, while for the ratio of 1:4, there are no differences in the trend of HSA-SH reactivity in the absence of or in the presence of glucose. The presence of glucose showed a positive effect while increasing the HSA:SA ratio showed a negative effect on the recovery of HSA-SH reactivity.

These results could be explained by the fact that in the presence of glucose in the incubation mixture, two glucose molecules are located in the Sudlow I site [15,16,17]. The cyclic form of glucose is more buried in the site, while the open form is located at the site’s entrance [16,17]. A molecular dynamics simulation study reveals that forming the dimeric structure of two sugar molecules by hydrogen bonds between them is essential for tightly binding the sugar dimers in the large cavity of Sudlow site I [16]. Furthermore, the presence of two molecules of sugars (glucose or galactose) in Sudlow site I increases the flexibility of this binding site and Sudlow site II [17,41]. In addition, binding sugars in Sudlow site I seem to make Trp214 the most flexible compared to Trp 214 in FA-free HSA and saturated HSA with myristic acid (the molar ratio of HSA:FAs 1:7), when it is stiffer [17]. In HSA saturated with FAs, the Trp214 indole ring is rigid and initially aligned perpendicular to a backbone axis because myristic acid is bound to Sudlow site I. Binding two molecules of glucose instead of myristic acid in Sudlow site I causes the rotation of the Trp214 indole ring from a perpendicular into a parallel position to a backbone axis, the most favorable conformation for an indole ring [17]. According to these data and our results, we can conclude that the binding of glucose in Sudlow site I, which is in the proximity of Cys34, is essential for maintaining HSA-SH group reactivity. Maintaining HSA-SH group content and reactivity is crucial for other plasma proteins with free thiol groups in their structure. Without the protective role of the HSA-SH group, the thiol group of other proteins will be more exposed to oxidative and carbonyl stress, which could lead to the loss of their function.

According to the obtained results, the following mechanism by which glucose and SA participate in maintaining thiol group reactivity could be suggested. Upon releasing FA-free HSA from the hepatocytes in the circulation, two glucose molecules are located in Sudlow site I [15]. Their interactions with HSA subsequently lead to increased flexibility at Sudlow site II [17], where one of the two binding sites for FAs with high affinity is located [2]. Our results showed that fluorescence quenching constants (Ksv) of SA to HSA increased 1.5 and 1.3 times in the presence of 5 and 10 mM glucose, respectively, compared to Ksv values obtained without glucose (Table 1). The binding of FAs in Sudlow II leads to the forming of a hydrogen bond between asparagine (Asp) 451 and Lys195 at the interface between Sudlow sites I and II. This hydrogen bond helps to tether subdomains IIA and IIIA, forcing subdomains IIIB and IB to move apart [17]. Moving of subdomain IB causes conformation changes of HSA molecules in the proximity of Cys34, located in a shallow crevice ~10 Å on the top of the heart-shaped structure of HSA [19]. As particular features of the protein environment modulate the HSA-SH reactivity, the conformation changes of HSA could lead to the Cys34 sulfur alternating from buried to exposed conformations [42]. It is well documented that the HSA-SH reactivity increases when different FAs are bound to HSA [4]. Moreover, after binding FAs to Sudlow site II, Lys195 formed a hydrogen bond with Asp451, and it is not available for glycation by the open form of glucose located in Sudlow site I. Anguizola et al. [33] reported that Lys525 has consistently been found to be a major site of glycation within both in vivo and in vitro glycated HSA, although other glycation sites have been observed. In normal physiological conditions, some HSA molecules do not bear FAs [1], and these molecules are more prone to glycation in the position Lys195 [33]. During the absence of free FAs in circulation, the hydrogen bond between Lys195 and Asp451 occasionally occurred in HSA with bound glucose [17]. This finding agrees with our result that the increased AGE formation after 14 days of incubation in the presence of 5 mM glucose was observed only in the absence of SA, while in the presence of SA molecules, the level of formation of AGEs was the same as the control (Figure 3).

## 4. Materials and Methods

### 4.1. Chemicals and Instrumentation

All chemicals of analytical reagent grade were purchased from Sigma-Aldrich (Steinheim, Germany) and Merck (Darmstadt, Germany) unless otherwise noted. The 20% solution of HSA (96% purity, intended for clinical use) was purchased from Baxter, Vienna, Austria. Deionized water (18 MΩ·cm at 25 °C) (Millipore, Bedford, MA, USA) was used for all experiments.

### 4.2. Preparation of FA-Free HSA

A commercial 20% solution of HSA used in the experiments contains free FAs. For experimental purposes, HSA was defatted using a charcoal treatment described by Chen [43]. Briefly, the HSA solution (200 g/L) was diluted with distilled water to a 1 mM solution; after that, 35 mg of activated charcoal was added per 1 mL of HSA, and the pH of the mixture was adjusted at 3.0 with hydrochloric acid (HCl) (0.2 M). After stirring for 1 h in an ice bath, charcoal was removed by centrifugation for 20 min at 10000 g (Eppendorf^®^ Minispin^®^, Hamburg, Germany). The supernatant was collected, and pH was brought to pH 7.0 with 0.2 M sodium hydroxide (NaOH). In the next step, HSA was washed several times with 0.1 M phosphate buffer, pH 7.4, and concentrated to 1 mM by ultrafiltration using a centrifugation tube, Ultracel-30K device (Millipore, Bedford, MA, USA).

### 4.3. Preparation of Highly Reduced HSA

Commercial 20% solution of HSA contains a non-physiological, low concentration of reduced HSA-SH (around 20%). For experimental purposes, obtained FA-free HSA was reduced with dithiothreitol as described by Penezić [5]. Briefly, before the reduction, the content of the HSA-SH group was determined. Then, FA-free HSA in 0.1 M sodium phosphate buffer, pH 7.4, was mixed with dithiothreitol at a molar ratio of 1:1 (molar content of oxidized thiol group:dithiothreitol). After incubation for 1 h at 37 °C, dithiothreitol was washed away from HSA with 0.1 M sodium phosphate, pH 7.4, using an Ultracel-30K device (Millipore, Bedford, MA, USA). After this treatment, the HSA-SH content was 0.89–0.92 mol SH group per mol HSA. For the incubation of HSA with FAs and glucose experiments, HSA-SH content was adjusted to 0.55 mol -SH/mol HSA by mixing appropriate volumes of highly reduced FA-free HSA and FA-free HSA.

### 4.4. Preparation of FA-Bound HSA

Analytical grade stearic acid (SA) was dissolved in 99% ethanol (concentration 50 mM) and mixed with FA-free and reduced HSA (0.55 mol -SH/mol HSA) in 0.1 M sodium phosphate buffer, pH 7.4 at 0:1 (control), 1:1, 2:1 and 4:1 molar ratio FA/HSA. Final ethanol concentrations in the HSA solutions were less than 2%. The mixtures were incubated at 37 °C for 1 h and then centrifuged for 5 min at 10,000× *g* [4]. After that, in the sterile zone, SA-bound HSA and control (SA-free HSA) solutions were passed through the sterile nitrocellulose syringe filters with 0.2 μm pores in the sterile vials (50 mL).

### 4.5. In Vitro Glycation of SA-Bound and SA-Free HSA

Before in vitro glycation of SA-bound HSA and control HSA (SA-free HSA), all glass materials, plastics (tubes, tips, and vials), and stock solutions of glucose (52 mM) in milli-Q water were sterilized using a lab autoclave. Sterile solutions of SA-bound HSA (1:1, 2:1, and 4:1) and SA-free HSA in 0.1 M sodium phosphate buffer, pH 7.4 M, were mixed with the stock solutions of glucose in the molar ratios of HSA:glucose, which corresponds to the physiological concentrations of HSA (0.64 mM) and pathophysiological concentrations of glucose (5, 10, and 20 mM), respectively, in 50 mL vials in the sterile zone. After mixing, the solutions were aliquoted in sterile tubes and incubated for 14 days at a constant temperature of 37 °C. Before (0) and after 1, 2, 3, 4, 6, 7, 10, and 14 days of incubation, tubes were taken for the analysis.

### 4.6. Quantification of HSA and HSA-SH Group Content

A biuret assay was used to quantify HSA [44]. The content of the reduced HSA-SH group in the samples of HSA during its preparation for glycation and in vitro glycation was determined spectrophotometrically according to the modified Ellman method using DTNB reagent [45]. Briefly, the reaction mixture consists of 100 μL of 1 M Tris buffer pH 8.0, 100 μL of the sample, 100 μL of DTNB (2 mM), and water up to 1000 μL. After incubation at room temperature for 30 min, the absorbance at 412 nm against the sample and reagent blanks was measured using a spectrophotometer Shimadzu UV/ViS 1800 (Kyoto, Japan). The concentration of thiols was calculated using the molar extinction coefficient (14,150 M^−1^ cm^−1^). The obtained values were expressed in mol SH/mol HSA.

### 4.7. Determination of the Pseudo-First-Order Constant for the Reaction of HSA-SH with DTNB

The reaction kinetics were monitored spectrophotometrically using the modified Ellman method [6]. Before analysis, all solutions were pre-incubated at 37 °C. Briefly, a sample (100 μL) of prepared mixtures of SA-free HSA and SA-bound HSA (1:1, 2:1, and 4:1) with glucose (0, 5, 10, and 20 mM), respectively, was mixed with 1 M Tris buffer pH 8.0 (100 μL), water (700 μL), and DTNB reagent (100 μL). After mixing, the absorbance at 412 nm was recorded every 5 s for the first 90 s, then every 10 s for 270 s, and finally every 30 s for the remaining 30 min of total reaction time at 37 °C using a Shimadzu UV/VIS 1800 (Kyoto, Japan) spectrophotometer equipped with a thermostat bath. To achieve the pseudo-first-order kinetics, the concentration of DTNB in the reaction mixture represented a forty-fold excess compared to the HSA-SH levels. The values of the pseudo-first-order kinetic constant (k^−1^) were determined by fitting the natural logarithm of unreacted HSA-SH concentration vs. reaction time using the linear least squares model.

### 4.8. Fluorometric Determination of HSA Structure Changes during In Vitro Glycation

The intrinsic fluorescence of HSA and the fluorescent properties of AGEs were used to follow HSA structural changes during in vitro glycation. The fluorescence measurements were performed on a FluoroMax-4 Jobin Yvon (Horiba Scientific, Kyoto, Japan) spectrofluorometer with a 1.0 cm quartz cell and thermostat bath. Excitation wavelengths were 280 nm (Trp and Tyr excitation) and 335 nm (AGEs excitation), and emission spectra in the ranges between 290–420 and 350–500 nm, respectively, were recorded at 37 °C as counts per second on corresponding wavelength. The excitation and emission slit widths were set at 3.0 nm. The samples of HSA from the incubation mixture were daily diluted with 0.1 M sodium phosphate buffer pH 7.4 to 2 μM as the final concentrations of HSA. Each spectrum was the average of three scans. After the spectra for the buffer were corrected, the obtained spectra were normalized [6].

### 4.9. Determination of the Quenching Constant of HSA Trp214 with SA

Solutions of FA-free HSA in 0.1 M sodium phosphate buffer pH 7.4 in the absence or the presence of glucose (5, 10, and 20 mM) were poured into a 1.0 cm quartz cell. Small aliquots of 357 μM SA solutions in ethanol were added to 2.5 mL of 2 μM HSA solution. Thus, the final concentrations of SA were 0-8 μM. Emission spectra were recorded in the 300 to 420 nm range at 37 °C with an excitation wavelength of 295 nm using FluoroMax-4 Jobin Yvon (Horiba Scientific, Kyoto, Japan) spectrofluorometer.

After correction of fluorescence intensity for the inner-filter effect according to Equation (1), the quenching constants of HSA/SA complexes were determined using the Stern–Volmer Equation (2) [46]:Fc = Fu × 10^(Aex dex+Aem dem)/2^(1)
where Fu is the measured emission fluorescence intensity, Fc is the corrected fluorescence intensity that would be measured in the absence of any inner-filter effects, dex and dem are the cell path lengths in the excitation and emission direction (1 cm), and Aex and Aem are the absorbance values of the quencher measured at the excitation and peak emission wavelength (340 nm).
F_0_/F = 1 + *k*_q_ *τ*_0_ [Q] = 1 + Ksv [Q](2)
where F_0_ and F are the HSA fluorescence intensities at 340 nm before and after the addition of the quencher (SA), Ksv is the Stern–Volmer quenching constant, *k*_q_ stands for the fluorescence quenching rate constant, *τ*_0_ is the fluorescence lifetime of the fluorophore, and [Q] is the quencher concentration.

### 4.10. Statistical Analysis

The results are expressed as means ± SD from two experiments performed in triplicate. The statistical comparisons between different incubation mixtures within the same time point were tested using one-way ANOVA. Levene’s test for homogeneity of variances was applied, and accordingly, the Tukey-HSD or the Games–Howell test was used for the post hoc multiple comparisons. All statistical analyses and graphical representations of data were performed using Origin 9.0 and the IBM SPSS Statistics 20 statistical programs. Values at *p* < 0.05 were considered to be significant differences.

## 5. Conclusions

Taken together, HSA is not as vulnerable as it seems, and there are mechanisms by which it maintains its functions in pathological conditions. For the first time, it has been shown in in vitro relevant conditions (0-1-2-4 molar ratios of SA to HSA and 5–20 mM glucose, 37 °C, 14 days) that FAs and glucose participate together in the mechanism that maintains the critical role of HSA as a major extracellular antioxidant, which basically secures extracellular antioxidant homeostasis. FAs protect HSA from glycation and this phenomenon is dependent on their molar ratio to HSA, for HSA:SA, 1:4 the effect is the most prominent. The SA affinity to HSA increased significantly (1.5- and 1.3-fold, *p* < 0.01) in the presence of 5 and 10 mM glucose compared to the control. More importantly, it was shown for the first time that glucose exposure regulates the binding affinity of FAs to HSA. Although this study has a limitation, because only the influence of SA was examined, it nevertheless provides enough evidence that the joint effect of FAs and glucose must be considered in future experiments. These effects of FAs and glucose are significant when the glycation of HSA is examined, or the binding constants of different substances to HSA are determined. These results deepen the knowledge about the possible regulation of the antioxidant role of HSA in diabetes and other pathophysiological conditions and enable the design of future HSA-drug studies which in turn is important for clinicians when designing information-based treatments.

## Figures and Tables

**Figure 1 ijms-25-02335-f001:**
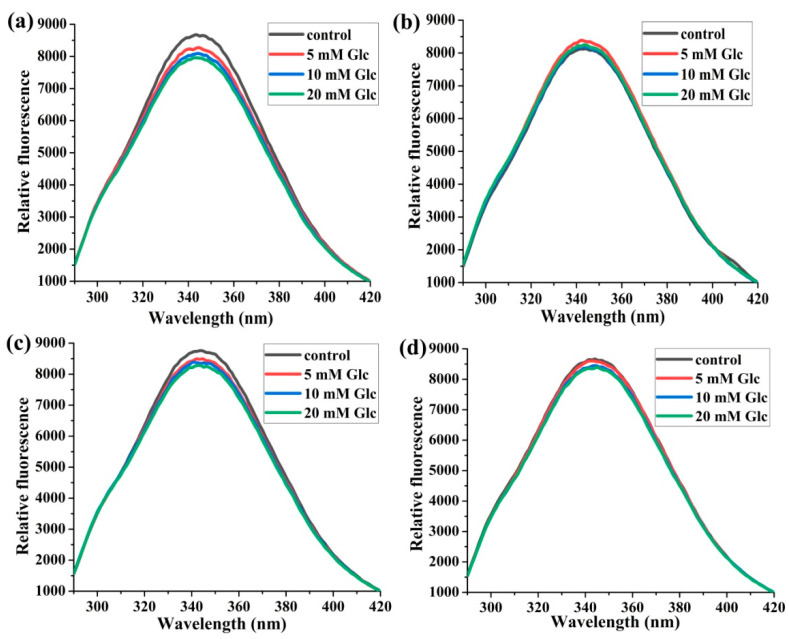
The effects of glucose (Glc) concentrations on the normalized spectra of Trp214 fluorescence. The incubation mixtures were prepared from HSA and SA in the molar ratios of HSA:SA 1:0 (**a**), 1:1 (**b**), 1:2 (**c**), and 1:4 (**d**) after 14 days of incubation at 37 °C in the absence of (control) or the presence of 5, 10, and 20 mM Glc. Before collecting spectra, HSA samples were diluted to the final concentration of 2 µM. Spectra were recorded at 37 °C after excitation at 280 nm.

**Figure 2 ijms-25-02335-f002:**
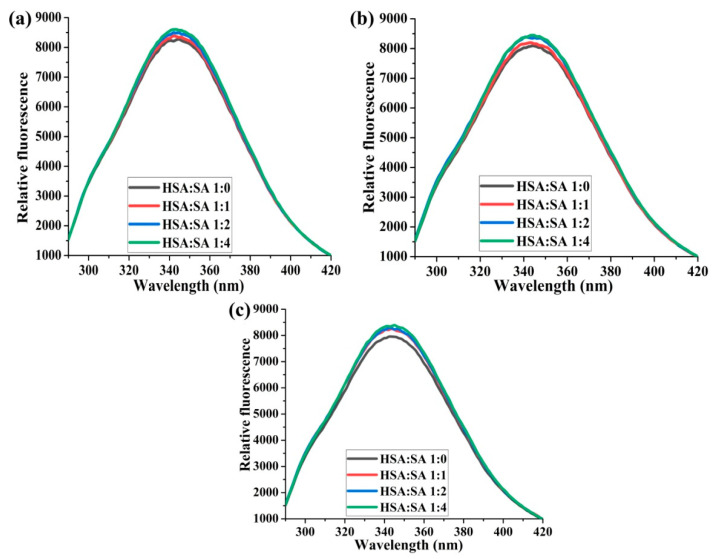
The effects of different HSA:SA molar ratios on the normalized Trp214 fluorescence spectral intensity, depicted by normal and pathological glucose (Glc) levels. The incubation mixtures were prepared from HSA and SA in the molar ratios of HSA:SA 1:0, 1:1, 1:2, and 1:4 after 14 days of incubation at 37 °C in the presence of 5 (**a**), 10 (**b**), and 20 (**c**) mM Glc. Before collecting spectra, HSA samples were diluted to the final concentration of 2 µM. Spectra were recorded at 37 °C after excitation at 280 nm.

**Figure 3 ijms-25-02335-f003:**
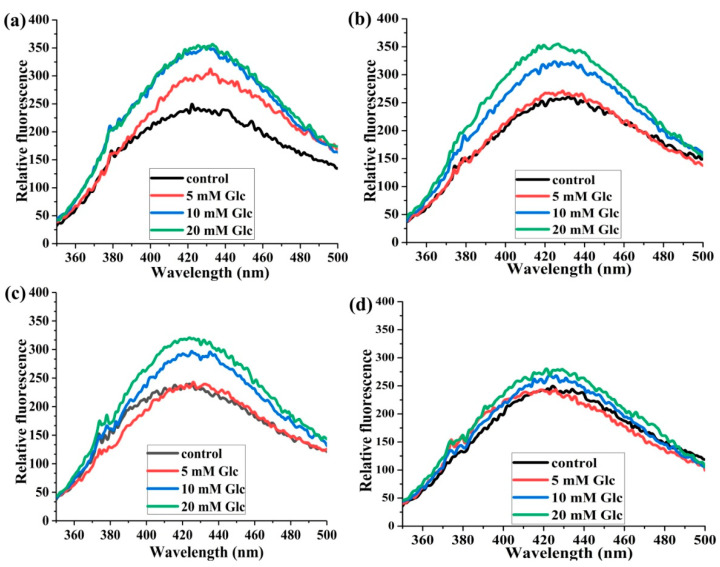
The effects of glucose (Glc) concentrations on the formation of AGEs. The incubation mixtures were prepared from HSA and SA in the molar ratios of HSA:SA 1:0 (**a**), 1:1 (**b**), 1:2 (**c**), and 1:4 (**d**) after 14 days of incubation at 37 °C in the absence of (control) or the presence of 5, 10, and 20 mM Glc. Before collecting spectra, HSA samples were diluted to the final concentration of 2 µM. Spectra were recorded at 37 °C after excitation at 335 nm.

**Figure 4 ijms-25-02335-f004:**
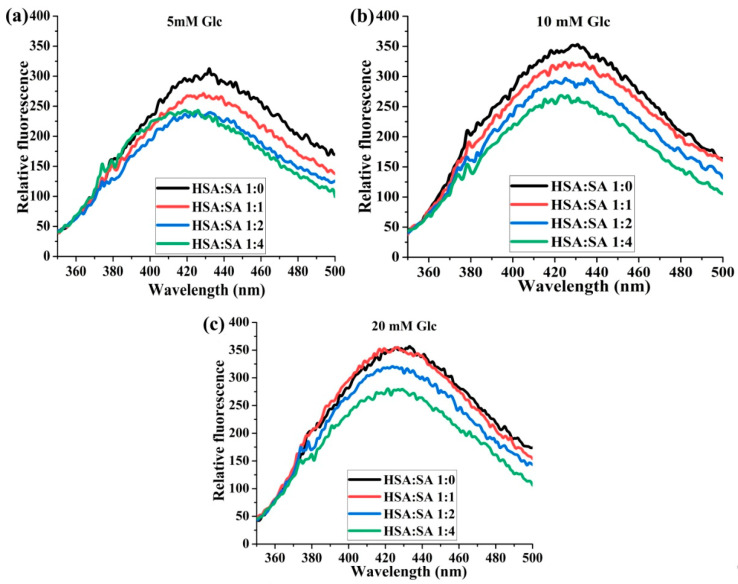
The effects of different molar ratios of HSA:SA on the formation of AGEs. The incubation mixtures were prepared from HSA and SA in the molar ratios of HSA:SA 1:0, 1:1, 1:2, and 1:4 after 14 days of incubation at 37 °C in the presence of 5 (**a**), 10 (**b**), and 20 (**c**) mM glucose (Glc). Before collecting spectra, HSA samples were diluted to the final concentration of 2 µM. Spectra were recorded at 37 °C after excitation at 335 nm.

**Figure 5 ijms-25-02335-f005:**
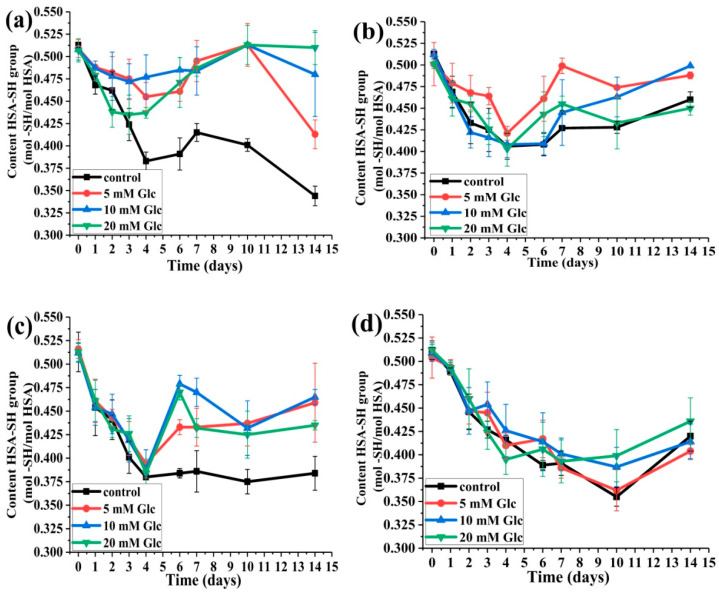
The effects of glucose (Glc) concentrations on the HSA-SH content determined by the Ellman method, during 14 days of incubation at 37 °C. The incubation mixtures were prepared from HSA and SA in the molar ratios of HSA:SA 1:0 (**a**), 1:1 (**b**), 1:2 (**c**), and 1:4 (**d**) in the absence (control) or the presence of 5, 10, and 20 mM Glc. Statistical differences between time points and mixtures are not shown for clarity reasons (Appendix A).

**Figure 6 ijms-25-02335-f006:**
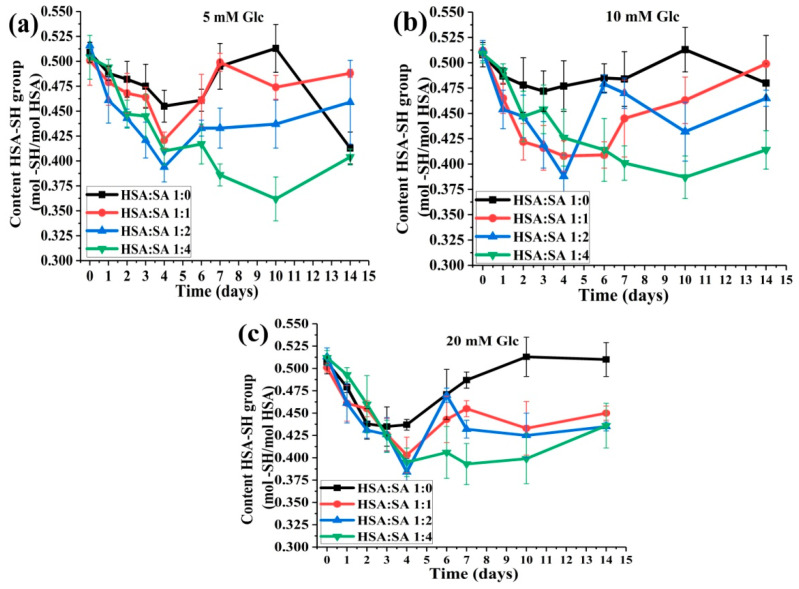
The effects of the HSA:SA molar ratios on the HSA-SH content, determined by the Ellman method, during 14 days of incubation at 37 °C. The incubation mixtures were prepared from HSA and SA in the molar ratios of HSA:SA 1:0, 1:1, 1:2, and 1:4 in the presence of 5 (**a**), 10 (**b**), and 20 (**c**) mM glucose (Glc). Statistical differences between time points and mixtures are not shown for reasons of clarity (Appendix A).

**Figure 7 ijms-25-02335-f007:**
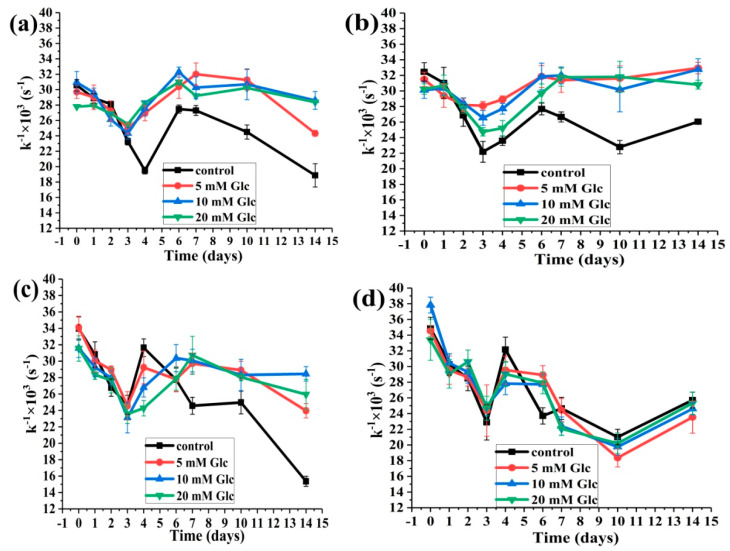
The effects of glucose (Glc) concentrations on the HSA-SH group reactivity (k^−1^ × 10^3^ s^−1^), determined by the modified Ellman method, during 14 days of incubation at 37 °C. The incubation mixtures were prepared from HSA and SA in the molar ratios of HSA:SA 1:0 (**a**), 1:1 (**b**), 1:2 (**c**), and 1:4 (**d**) in the absence (control) or the presence of 5, 10, and 20 mM Glc. Statistical differences between time points and mixtures are not shown for reasons of clarity (Appendix A).

**Figure 8 ijms-25-02335-f008:**
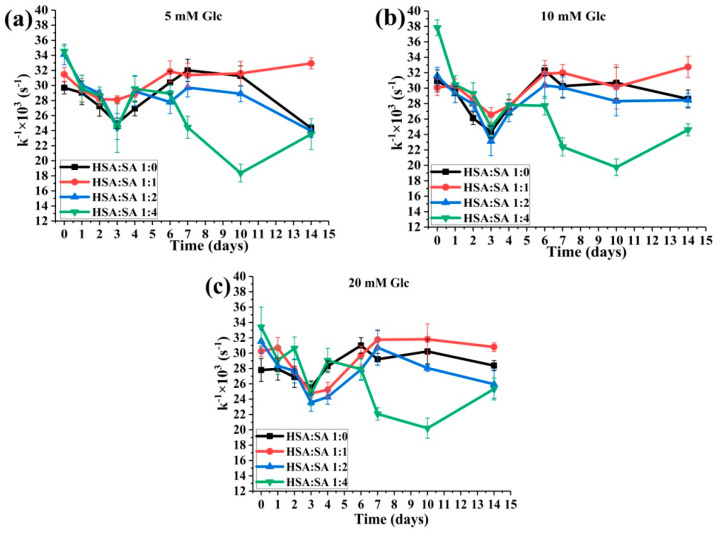
The effects of the HSA:SA molar ratios on the HSA-SH reactivity (k^−1^ × 10^3^ s^−1^), determined by the modified Ellman method, during 14 days of incubation at 37 °C. The incubation mixtures were prepared from HSA and SA in the molar ratios of HSA:SA 1:0, 1:1, 1:2, and 1:4 in the presence of 5 (**a**), 10 (**b**), and 20 (**c**) mM glucose (Glc). Statistical differences between time points and mixtures are not shown for reasons of clarity (Appendix A).

**Table 1 ijms-25-02335-t001:** Stern–Volmer constants (Ksv) of binding SA to HSA in the absence (control) or the presence of 5, 10, and 20 mM glucose. The experiment was performed two times at 37 °C.

	Ksv × 10^4^ (mol ^−1^ × L)			
	Control	HSA + 5 mM Glc	HSA + 10 mM Glc	HSA + 20 mM Glc
Experiment 1	1.28	1.98	1.60	1.42
Experiment 2	1.24	1.83	1.65	1.14
Mean ± SD	1.26 ± 0.03	1.91 ± 0.10 *	1.63 ± 0.04 **	1.28 ± 0.19 ^+^

* *p* < 0.01 compared to the control value; ** *p* < 0.05 compared to the control value; ^+^ *p* < 0.05 compared to HSA+ 5 mM glucose (Glc).

## Data Availability

Data are contained within the article.

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
