# Peer review of "The Thiol Group Reactivity and the Antioxidant Property of Human Serum Albumin Are Controlled by the Joint Action of Fatty Acids and Glucose Binding"

_ijms, 2024, doi:10.3390/ijms25042335_

Round 1

Reviewer 1 Report

Comments and Suggestions for Authors

The manuscript “Albumin thiol group content and reactivity are under control of joint action of fatty acids and glucose binding.” was written by Tamara Uzelacand and coauthors. The function changes of HSA in pathological conditions is an important issue. This protein is one of the abundant blood plasma proteins rich in binding sites directed to small molecules: drugs, fatty acids, and metal ions. The article has a good overall impression. It has an appropriate sectional structure for research work, which is presented concisely and clearly. The introduction provides a good background. The discussion of research results is justified and supported by experiments. However, there are a few points that require correction, listed below:

1.      In Table 1 the Ksv needs a unit, please fill it up.

2.      Also, the fluorescence data, and more specifically regarding the constant (KSV), require a broader discussion and comparison with other literature values.

3.      The resolution of the figures should be higher to make the data easier to read. I suggest increasing the font on the axes and the legend.

4.      The fluorescence spectra to monitor the effect of glucose concentration and different molar ratios of HSA:SA on the formation of AGEs possess a high signal-to-noise ratio. What may be the cause of this and can it be avoided or eliminated in some way?

5.      Last point is only a comment. The fluorescence spectroscopic method is commonly used to control structural changes in proteins. This is possible due to the presence of amino acids such as Phe, Trp, Tyr in the sequence. Although it is obvious that Trp shows the greatest fluorescence. This is the basis for research on this type of interaction. However, other spectroscopic methods monitoring the secondary structure of a protein could also be used, e.g. circular dichroism (CD) or FT-IR. Please reconsider the expand your research in the future.

6.      Please read carefully the whole article with attention to editorial and typos bags. In my opinion, they are small and do not diminish work, but should be corrected.

Comments on the Quality of English Language

 The English language is fine, only small editorial bugs require correction. The manuscript is written at the scientific level but clear and understandable.    

Reviewer 2 Report

Comments and Suggestions for Authors

The manuscript presents an experimental study of the human serum albumin in the presence of glucose and fatty acids. Although the topic sounds scientifically, the presentation of the material seems  more like description of the accumulated data rather than its analysis. 

It is generally not clear what cab be concluded from experiment except from "These results are physiologically relevant because they explain different phenomena observed in pathophysiological conditions. Thus, I suggest the authors to at least speculate on the origin of the results obtained.

Comments on the Quality of English Language

English language is ok

Reviewer 3 Report

Comments and Suggestions for Authors

This article is well written on (This article is well written on (Albumin thiol group content and reactivity are under control of 2 joint action of fatty acids and glucose binding), however here are a few suggestions that will improve the quality of the manuscript if followed by the authors

1.      The abstract is fine but it needs to be a bit more focused on the aim and methodology sentences, the conclusion part of the abstract should be improved. There is no statistical analysis and no conclusive line is in the abstract section. Please revise it and make its sequence.

2.      Please update the introduction  section and briefly explain about the binding of glucose and fatty acids along with the disease. It would be much better if you maintain the sequence of the introduction.

3.      Please add rationale specifically reasoning of the study in the end of introduction section.

4.      Results are somewhat confusing it would be much better if you could provide tabulated data either in the manuscript or supplementary files.

5.      Justification and reasoning is missing in the discussion section, authors should must provide point by point justification of results and the reasoning.

6.      Please reduce the discussion part and focus on major achievements.

7.      Please provide references in each section of methodology.

8.      Find a more recent reference than “Peters, T. All About Albumin: Biochemistry, Genetics and Medical Applications, 1st ed.; Academic Press: San Diego, CA, 1996 591”

9.      “Evans, T.W. Review article: albumin as a drug–biological effects of albumin unrelated to oncotic pressure. Aliment. Pharmacol. 592 Ther. 2002, 16, 6–11. doi: 10.1046/j.1365-2036.2002.00190.x” it is very old reference. Try to find recent.

10.  Find a more recent reference than  “Kragh-Hansen, U. Molecular aspects of ligand binding to serum albumin. Pharmacol. Rev. 1981, 33, 17–53. doi:003l-6997/81/3301- 596 00l7”

11.  What is the aim of the manuscript? What new will it present? Will it attempt to answer any outstanding questions? If so, which ones? Can you write a para in the introduction to grab the reader’s attention?

12.  The writing and presentation is good in some places but awkward and cumbersome in others. A sentence-by-sentence copy-editing by someone with the appropriate expertise is needed.

13.  Adjust paragraph length to align with journal’s formatting guidelines.

14.  Please mention the full form first and then abbreviations in the following text of the manuscript.

15.  Improve grammatical mistakes in the whole manuscript.

16.  Conclusions: Enhance the conclusion by adding numerical values with mean findings. Additionally, Do your findings suggest recommendations for clinicians?

17.  Improve references according to journal requirements.

Comments on the Quality of English Language

 English very difficult to understand/incomprehensible

Round 2

Reviewer 2 Report

Comments and Suggestions for Authors

Although the manuscript is improved , I fill that it would better suit some other journal, but not IJMS.

Comments on the Quality of English Language

Ok

Reviewer 3 Report

Comments and Suggestions for Authors

It could be accepted in present form

Comments on the Quality of English Language

It looks fine